

# Clinical value of serum DJ-1 in lung adenocarcinoma

Lin Wang[*], Li Wei[*], Shuxian Miao and Wei Zhang

Department of Laboratory Medicine, the First Affiliated Hospital of Nanjing Medical University, Nanjing, China
Branch of National Clinical Research Center for Laboratory Medicine, Nanjing, China
[*] These authors contributed equally to this work.

## ABSTRACT

**Objective**. DJ-1 is an oncoprotein secreted by cancer cells. However, the physiological and pathological significance of DJ-1 secretion is not clearly understood. This study investigated the clinical value of serum DJ-1 in lung adenocarcinoma (LUAD).

**Methods**. The study involved 224 LUAD patients, 110 patients with benign pulmonary disease and 100 healthy controls from the First Affiliated Hospital of Nanjing Medical University. We detected the expression of DJ-1 in lung cell lines *in vitro*. Meanwhile, serum concentrations of DJ-1, carcinoembryonic antigen (CEA), neuron-specific enolase (NSE), and cytokeratin 19 fragment (CYFRA21-1) were measured. The diagnostic performance of LUAD was obtained using receiver operating characteristic (ROC) curves. Kaplan–Meier, univariate and multivariate Cox regression analyses were performed for progression-free survival (PFS).

**Results**. DJ-1 was highly expressed in LUAD cell lines. Serum DJ-1 levels were significantly higher in the LUAD group compared to the benign pulmonary disease group (5.04 *vs.* 3.66 ng/mL, $P < 0.001$) and healthy controls (5.04 *vs.* 3.51 ng/mL, $P < 0.001$). DJ-1 levels were associated with gender ($P = 0.002$), smoking history ($P = 0.042$) and lymph node metastasis ($P = 0.040$). ROC curve analysis of DJ-1 revealed an area under the curve (AUC) of 0.758 (95% CI [0.714–0.803], $P < 0.001$) with a sensitivity of 63.8% and specificity of 78.6% at a cutoff value of 4.62 ng/mL for the detection of LUAD. Univariate and multivariate analyses confirmed that the preoperative serum DJ-1 level, tumor stage and smoking history were independent prognostic factors of PFS.

**Conclusion**. Our study is the first to explore the clinical value of serum DJ-1 in LUAD comprehensively. Serum DJ-1 could be a potential diagnostic and prognostic biomarker for LUAD.

## INTRODUCTION

Lung cancer is the most commonly diagnosed cancer and remains a major reason for cancer-related deaths worldwide with an estimated 2.2 million new cases (11.4%) and 1.8 million deaths (18%) in 2020 (*Sung et al., 2021*). A large proportion of lung cancer patients are diagnosed at an advanced stage and the 5-year survival rate is approximately 23% (*Siegel et al., 2023*). The 2020 global cancer statistics reported by the International Agency for

Corresponding author
Wei Zhang, zhang_wei@njmu.edu.cn

Research on Cancer revealed that an estimated 820,000 new lung cancer diagnoses and 715,000 lung cancer-related deaths occurred in China in 2020 (*Sung et al., 2021*; *Cao et al., 2021*). Lung cancer can be divided into small cell lung cancer (SCLC) and non-small cell lung cancer (NSCLC) and NSCLC represents approximately 85%. Lung adenocarcinoma (LUAD) is the most common subtype of NSCLC (*Davidson, Gazdar & Clarke, 2013*). LUAD is concealed at the onset by rapid development and poor prognosis. Traditional serum tumor markers, such as carcinoembryonic antigen (CEA), neuron-specific enolase (NSE), and cytokeratin 19 fragment (CYFRA21-1), have been used for detecting LUAD for a long time. However, they showed insufficient specificity or sensitivity. Therefore, efficient tumor molecular biomarkers for early diagnosis and prognosis are essential for improving patient survival.

DJ-1 was initially identified as an oncogene in 1997 and it is a 189 amino acid protein that can transform mouse NIH3T3 cells in cooperation with the activated ras gene (*Nagakubo et al., 1997*). Subsequently, it was named Parkinson's disease (PD)-associated protein 7 (PARK7) in 2003 as it is able to protect neurons from oxidative stress (*Bonifati et al., 2003*). *Waragai et al. (2006)* found a higher level of DJ-1 in the cerebrospinal fluids of sporadic Parkinson's disease in 2006. DJ-1 is present in various cells and has multiple functions in numerous physiological and pathophysiological processes, such as cell proliferation and growth, apoptosis, gene transcription, and cellular defense against oxidative stress (*Parsanejad et al., 2014*; *Bonilha et al., 2017*; *Liu et al., 2019*; *Meiser et al., 2016*). DJ-1 is highly expressed in different types of cancer with poor prognosis including lung, breast, cervical, brain, endometrial, pancreatic and thyroid cancer (*Han et al., 2017*; *Wang et al., 2020*; *Schabath & Cote, 2019*; *Kawate, Tsuchiya & Iwaya, 2017*). DJ-1 plays functional roles in cancer progression. For example, as a positive regulator, DJ-1 participates the Androgen Receptor (AR)-signaling pathway (*Niki et al., 2003*). DJ-1 inhibits apoptosis by inducing surviving expression (*Shen et al., 2011*). DJ-1also modulates oncoproteins and tumor suppressors expression (*Jin, 2020*). DJ-1 can be secreted into the blood by cancer cells and serum DJ-1 is reported to be elevated in pancreatic cancer (*He et al., 2011*), which suggest that serum DJ-1 might be used as a potential biomarker reflecting tumor occurrence and development. However, the clinical significance of DJ-1 in the diagnosis and prognosis of LUAD remains unclear. In this study, we evaluated the clinical value of serum DJ-1 in LUAD.

## MATERIALS AND METHODS

### Study population

This retrospective study enrolled 224 LUAD patients, 110 patients with benign pulmonary disease and 100 healthy controls from the First Affiliated Hospital of Nanjing Medical University between January 2016 and July 2017. The inclusion criteria were as follows: (1) LUADs were confirmed by pathology and (2) complete clinical data. The exclusion criteria were as follows: (1) patients had a previous history of other cancers or Parkinson's disease and (2) received any treatment before surgery. During the same period, 110 patients with benign lung disorders were included as the benign pulmonary disease

group. Healthy controls were recruited at the Health Management Center and excluded individuals with a history of other cancers and any lung diseases. During the follow-up period, all LUAD patients underwent chest CT and serum tumor markers every 6 to 8 weeks to assess the tumor progression. All LUAD patients were followed up until September 2022. Progression-free survival (PFS) was defined as the time to progression or death using the Response Evaluation Criteria in Solid Tumors criteria (RECIST) v1.1 criteria. This study was approved by the Institutional Ethics Committee of the First Affiliated Hospital of Nanjing Medical University (2022-SR-621), and informed consent was specifically waived by the ethics committee.

## Cell culture

Human LUAD cell lines (SPC-A1, A549), human bronchial epithelial cell line (HBE) were obtained from the Chinese Academy of Sciences, China. All the cells were cultured in RPMI1640 medium (Gibco, Carlsbad, CA, USA) reconstituted with 1% penicillin-streptomycin (Gibco, Carlsbad, CA, USA) and 10% fetal bovine serum (Gibco, Carlsbad, CA, USA) at 37 °C in a humidified atmosphere with 5% $CO_2$.

## Reverse transcription quantitative polymerase chain reaction

Total RNA was extracted from lung cell lines with TRIzol reagent (Invitrogen, Waltham, MA, USA). A PrimeScript RT Reagent Kit (TaKaRa, Shiga, Japan) was used for cDNA systhesis. Quantitative polymerase chain reaction (qPCR) was performed on a 7500 Real-Time PCR System (Applied Biosystems, Waltham, MA, USA). The relative DJ-1 expression compared with β-actin was calculated using the $2-\Delta\Delta CT$ method. The primers sequences were listed in Supplemental Files.

## Serum marker detection

Serum from all participants was collected on the second day of admission for enzyme-linked immunosorbent assay (ELISA) analysis. After venous blood collection, the blood samples were centrifuged at 4,000 rpm for 10 min, and then the serum was transferred into Eppendorf tubes and stored at −70 °C until analysis.

DJ-1 concentrations were analyzed by ELISA with commercial Human Park7/DJ-1 ELISA kits (R&D, Minneapolis, MN, USA) according to the manufacturer's instructions. The limit of detection was 6.25 pg/mL, each sample was examined in duplicate, and the mean values were used in subsequent statistical analyses.

Serum levels of CEA, CYFRA21-1 and NSE were measured on a Cobas e602 analyzer with Elecsys kits (Roche Diagnostics Corp., Indianapolis, IN, USA). These assays utilize the electrochemiluminescence immunoassay (ECLIA) method, and the unit of measurement is defined in nanograms per milliliter (ng/mL).

## Statistical analysis

The statistical analyses were performed with SPSS software (version 22.0). Continuous data were described using the median and range with the Mann–Whitney U test or Kruskal–Wallis test for nonparametric comparison. Receiver operating characteristic (ROC) curves were used to calculate the diagnostic performance. A *P* value of 0.05 was considered

statistically significant. Cox proportional hazards regression model was used to determine the independent predictive factors of PFS. $P < 0.05$ was used to select the variables from the univariate analysis to enter multivariate model. Kaplan–Meier analysis and log-rank test was used to compare the PFS of different risk groups, $P < 0.05$ was statistically significant. Bayesian shrinkage prior models were used as alternative approaches to validate the data in this study (*Bhattacharyya et al., 2022*).

## RESULTS

### The expressions of DJ-1 in lung cell lines

We detected DJ-1 concentration in cell culture supernatant by ELISA and mRNA by RT-PCR. The expressions of DJ-1 in lung cell lines are shown in Fig. 1. Both the DJ-1 levels of cellular supernatant and relative DJ-1 mRNA expressions were higher in LUAD cell lines (SPC-A1, A549), compared to HBE cell line ($P < 0.001$).

### Upregulation of serum DJ-1 levels in LUAD patients

The characteristics of LUAD patients and control groups are described in Table 1. There were no significant differences in age or sex. The distribution of serum DJ-1 levels in the LUAD group, benign pulmonary disease group and healthy control group are shown in Fig. 2; the median serum DJ-1 levels were 5.04 ng/mL, 3.66 ng/mL and 3.51 ng/mL, respectively. Serum DJ-1 levels were significantly higher in the LUAD group than in the benign pulmonary disease group ($P < 0.001$) and healthy control group ($P < 0.001$).

### Associations of serum DJ-1 levels with clinicopathological parameters of LUAD

The serum DJ-1 levels in groups with different clinicopathological parameters are shown in Table 2. Serum DJ-1 in male patients was significantly higher than in female patients ($P = 0.002$). Furthermore, serum DJ-1 expression was significantly correlated with smoking history ($P = 0.042$) and lymph node metastasis ($P = 0.040$). No differences were observed in LUAD patients grouped by age, tumor size, tumor number, tumor stage, distant metastasis, a history of diabetes and hypertension.

### Diagnostic performance of DJ-1, CEA, CYFRA21-1 and NSE in LUAD

To evaluate the diagnostic performance of DJ-1, CEA, CYFRA21-1 and NSE in LUAD, we performed a ROC analysis (Fig. 3). Serum DJ-1 showed the best diagnostic value among all markers for discriminating LUAD *versus* the controls. The AUC of DJ-1 was 0.758 (95% CI [0.714–0.803], $P < 0.001$) with a sensitivity of 63.8% and a specificity of 78.6% at a cutoff value of 4.62 ng/mL. The AUCs for CEA, CYFRA21-1 and NSE were 0.579 (95% CI [0.526–0.633], $P = 0.004$), 0.496 (95% CI [0.442–0.551], $P = 0.896$) and 0.647 (95% CI [0.596–0.699], $P < 0.001$), respectively. The sensitivity, specificity, positive predictive value (PPV) and negative predictive value (NPV) of the four markers in detecting LUAD are shown in Table 3.

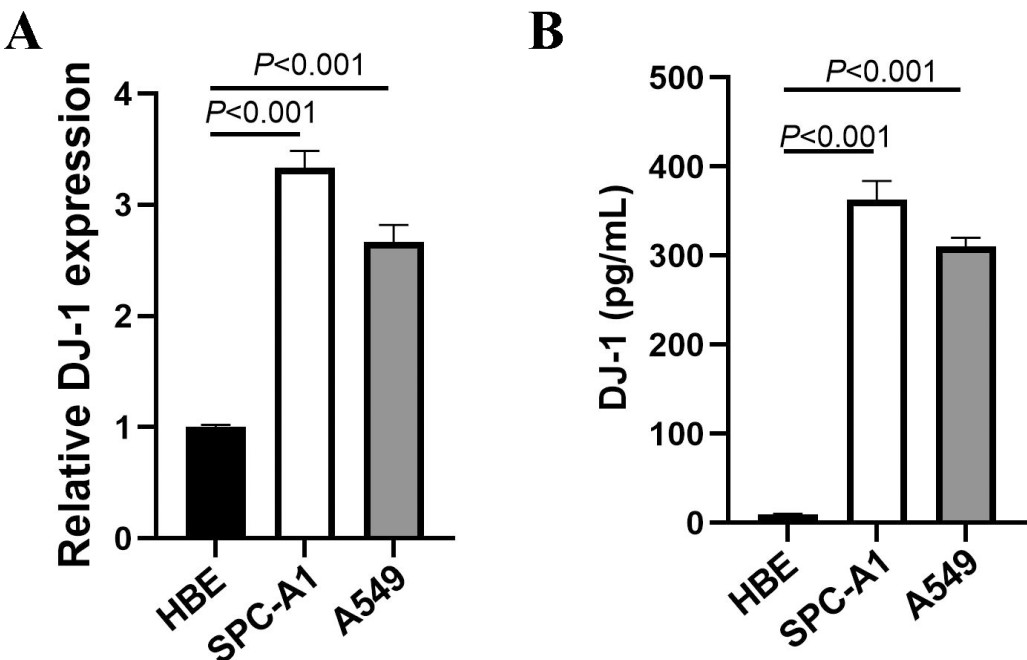

**Figure 1** **Expressions of DJ-1 in lung cell lines.** (A) DJ-1 mRNA expressions in lung cell lines. (B) ELISA results of DJ-1 expression in supernatant of lung cell lines.

**Table 1** **Demographic and clinical features of the study populations.**

| Characteristic | Lung adenocarcinoma (n = 224) | Benign pulmonary disease (n = 110) | Healthy controls (n = 100) |
|---|---|---|---|
| Age (years) | | | |
|   Median | 59 | 58 | 57 |
|   Range | 24–87 | 20–86 | 22–86 |
| Gender (n, %) | | | |
|   Male | 92 (41.1) | 45 (41.0) | 40 (40.0) |
|   Female | 132 (58.9) | 65 (59.0) | 60 (60.0) |
| Smoking (n, %) | | | |
|   Yes | 37 (16.5) | 20 (18.2) | 15 (15.0) |
|   No | 187 (83.5) | 90 (81.8) | 85 (85.0) |

## Serum DJ-1 is significantly and independently associated with PFS in LUAD

All LUAD cases were had a median follow-up period of 50.0 months. The 1-, 3-, and 5-year progression-free survival rates were 90.1%, 78.9% and 70.1%, respectively. The ROC curves of the four markers for predicting PFS in LUAD patients are shown in Fig. 4. According to ROC analysis, the AUC of DJ-1 for predicting PFS was 0.726 (95% CI [0.658–0.794], $P < 0.001$) with a cutoff value of 4.99 ng/mL. In addition, the AUCs of CEA, CYFRA21-1 and NSE were 0.566 (95% CI [0.483–0.648], $P = 0.134$), 0.459 (95% CI [0.371–0.564],

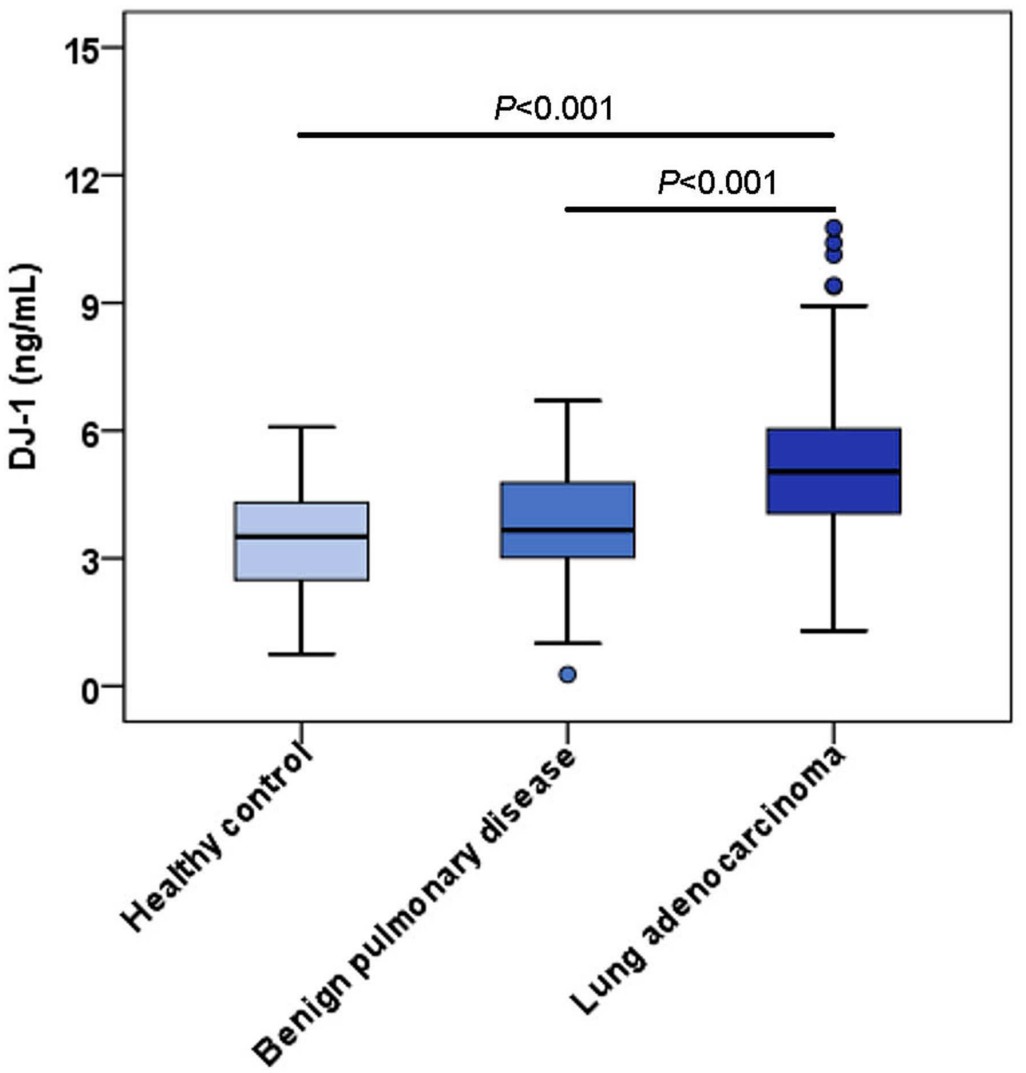

**Figure 2  Serum levels of DJ-1 among the controls and LUAD cases.** Each box refers to the 25th and 75th percentile values with a line indicating median levels, whereas the 95% confidence interval extends beyond the box. Points outside the 95% confidence intervals are outliers.

$P = 0.345$) and 0.639 (95% CI [0.559–0.719], $P = 0.002$), respectively. A Kaplan–Meier analysis revealed that patients with high DJ-1 levels displayed worse median PFS than those with low DJ-1 levels (32.5 months *vs.* 58.0 months, $P < 0.001$, Fig. 5). The results of the univariate and multivariate analyses for PFS are shown in Table 4. In a univariate analysis, PFS was significantly associated with gender (HR 0.519, 95% CI [0.311–0.868], $P = 0.012$), tumor size (HR 2.039, 95% CI [1.210–3.435], $P = 0.007$), tumor stage (HR 3.255, 95% CI [1.894–5.592], $P < 0.001$), lymph node metastasis (HR 2.393, 95% CI [1.291–4.435], $P = 0.006$), differentiation (moderate *vs.* well, HR 2.321, 95% CI [1.078–4.998], $P = 0.031$; poor *vs.* well, HR 3.422, 95% CI [1.601–7.312], $P < 0.001$), smoking (HR 2.497, 95% CI [1.417–4.402], $P = 0.002$) and high DJ-1 (HR 5.696, 95% CI [2.933–11.059], $P < 0.001$).

**Table 2  Correlation between serum DJ-1 levels and clinicopathological characteristics of 224 LUAD patients.**

| Characteristics | n | DJ-1 (ng/mL) | | P value |
|---|---|---|---|---|
| | | Median | Range | |
| **Gender** | | | | |
| Male | 92 | 5.44 | 1.33–12.58 | 0.002 |
| Female | 132 | 4.78 | 1.30–12.39 | |
| **Age (years)** | | | | |
| ≤60 | 121 | 4.87 | 1.30–12.51 | 0.504 |
| >60 | 103 | 5.20 | 1.39–12.58 | |
| **Tumor size (cm)** | | | | |
| ≤2 | 159 | 5.03 | 1.33–12.58 | 0.892 |
| >2 | 65 | 5.05 | 1.30–12.39 | |
| **Tumor number** | | | | |
| Single | 189 | 5.05 | 1.30–12.58 | 0.209 |
| Multiple | 35 | 4.62 | 2.63–12.51 | |
| **Tumor stage** | | | | |
| I | 183 | 5.03 | 1.30–12.58 | 0.932 |
| II–IV | 41 | 5.11 | 2.08–10.41 | |
| **Lymph node metastasis** | | | | |
| Yes | 30 | 5.56 | 2.90–12.51 | 0.040 |
| No | 194 | 4.96 | 1.30–12.58 | |
| **Distant metastasis** | | | | |
| Yes | 8 | 5.58 | 3.95–9.41 | 0.304 |
| No | 216 | 5.03 | 1.30–12.58 | |
| **Differentiation** | | | | |
| Well | 66 | 5.29 | 2.63–9.38 | 0.426 |
| Moderate | 88 | 4.82 | 1.33–12.51 | |
| Poor | 70 | 5.05 | 1.30–12.58 | |
| **Smoking** | | | | |
| Yes | 37 | 5.30 | 1.39–12.58 | 0.042 |
| No | 187 | 4.96 | 1.30–12.39 | |
| **Hypertension** | | | | |
| Yes | 54 | 5.00 | 2.13–12.39 | 0.889 |
| No | 170 | 5.04 | 1.30–12.58 | |
| **Diabetes mellitus** | | | | |
| Yes | 24 | 5.76 | 1.39–8.52 | 0.052 |
| No | 200 | 4.99 | 1.30–12.58 | |

Multivariate analysis demonstrated that tumor stage (HR 3.089, 95% CI [1.785–5.346], $P < 0.001$), smoking (HR 1.820, 95% CI [1.021–3.244], $P = 0.042$) and high DJ-1 (HR 5.298, 95% CI [2.697–10.406], $P < 0.001$) were independent prognostic factors of PFS.

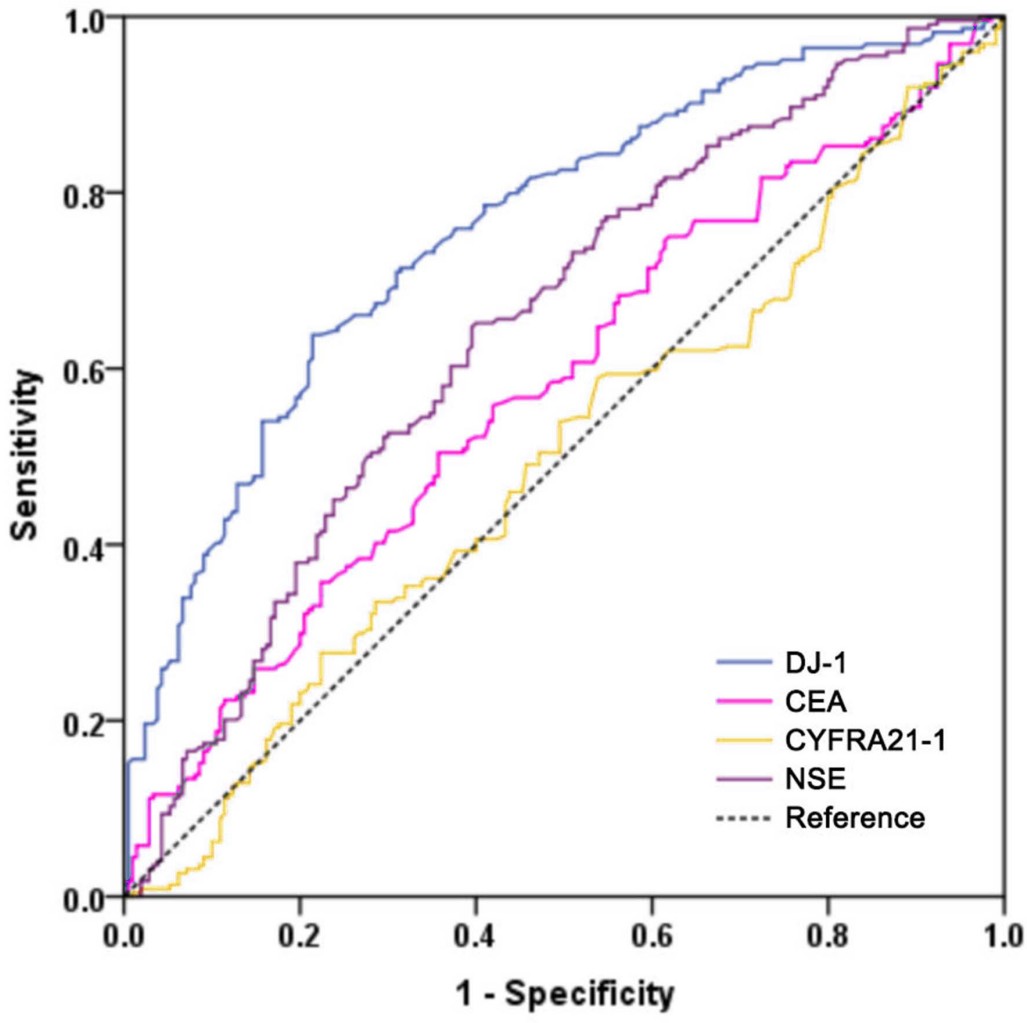

**Figure 3** Receiver operating characteristic curves of DJ-1, CEA, CYFRA21-1 and NSE for the diagnosis of LUAD in all patients.

**Table 3** A diagnostic performance of four biomarkers in detecting patients with LUAD.

| Biomarkers | P | AUC | 95% CI | Cut-off value | Sensitivity (%) | Specificity (%) | PPV (%) | NPV (%) |
|---|---|---|---|---|---|---|---|---|
| DJ-1 | <0.001 | 0.758 | 0.714–0.803 | 4.62 | 63.8 | 78.6 | 76.1 | 67.1 |
| CEA | 0.004 | 0.579 | 0.526–0.633 | 2.38 | 50.4 | 64.3 | 60.1 | 54.9 |
| CYFRA21-1 | 0.896 | 0.496 | 0.442–0.551 | 1.79 | 58.9 | 46.2 | 53.9 | 51.3 |
| NSE | <0.001 | 0.647 | 0.596–0.699 | 13.87 | 58.9 | 60.0 | 61.1 | 57.8 |

Notes.
Abbreviations: AUC, areas under the curve; PPV, positive predictive value; NPV, negative predictive value.

## DISCUSSION

LUAD represents one of the most common and aggressive human lung malignancies in the world and is associated with a poor prognosis. Early diagnosis, which gives patients

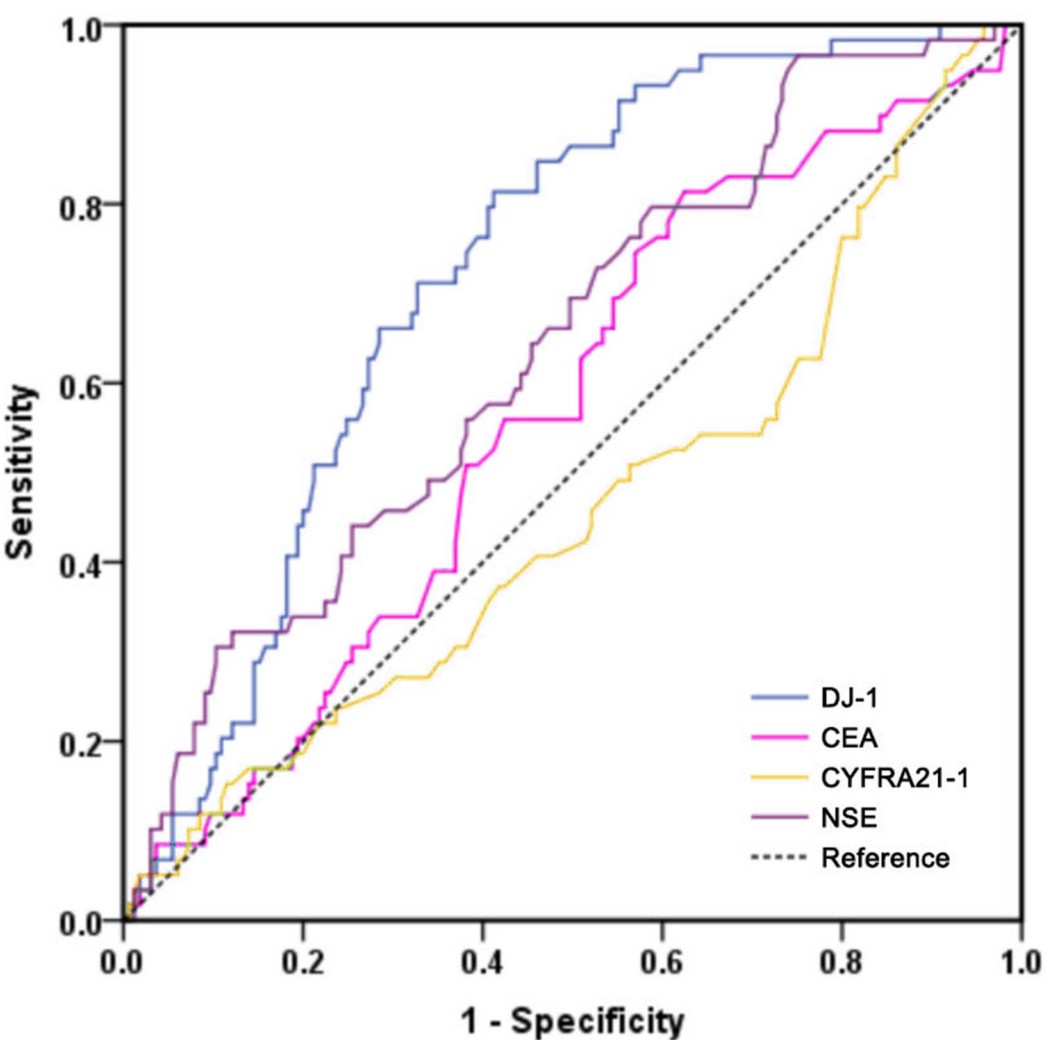

**Figure 4** ROC curves of DJ-1, CEA, CYFRA21-1 and NSE for predicting PFS in patients with LUAD.

the chance to receive efficient therapy in the early stage, is therefore highly desirable, especially noninvasive diagnostic methods such as serological markers. Our study is the first to investigate the clinical value of serum DJ-1 in both the diagnosis and prognosis of LUAD. Compared to other clinical specimens, serum is easier to obtain and so serum DJ-1 may be used as a routine laboratory parameter.

In this study, we first detected the expression of DJ-1 in lung cell lines *in vitro*, then we analyzed serum concentrations of DJ-1 in LUAD patients, patients with benign pulmonary disease and healthy controls. Consequently, DJ-1 expressions were higher in LUAD cell lines than HBE cell line. serum DJ-1 was significantly increased in the LUAD group. Furthermore, we observed that DJ-1 was associated with sex, smoking history and lymph node metastasis. The ROC curve analysis of DJ-1 revealed an AUC of 0.758 with a sensitivity of 63.8% and a specificity of 78.6% at a cutoff value of 4.62 ng/mL for the detection of

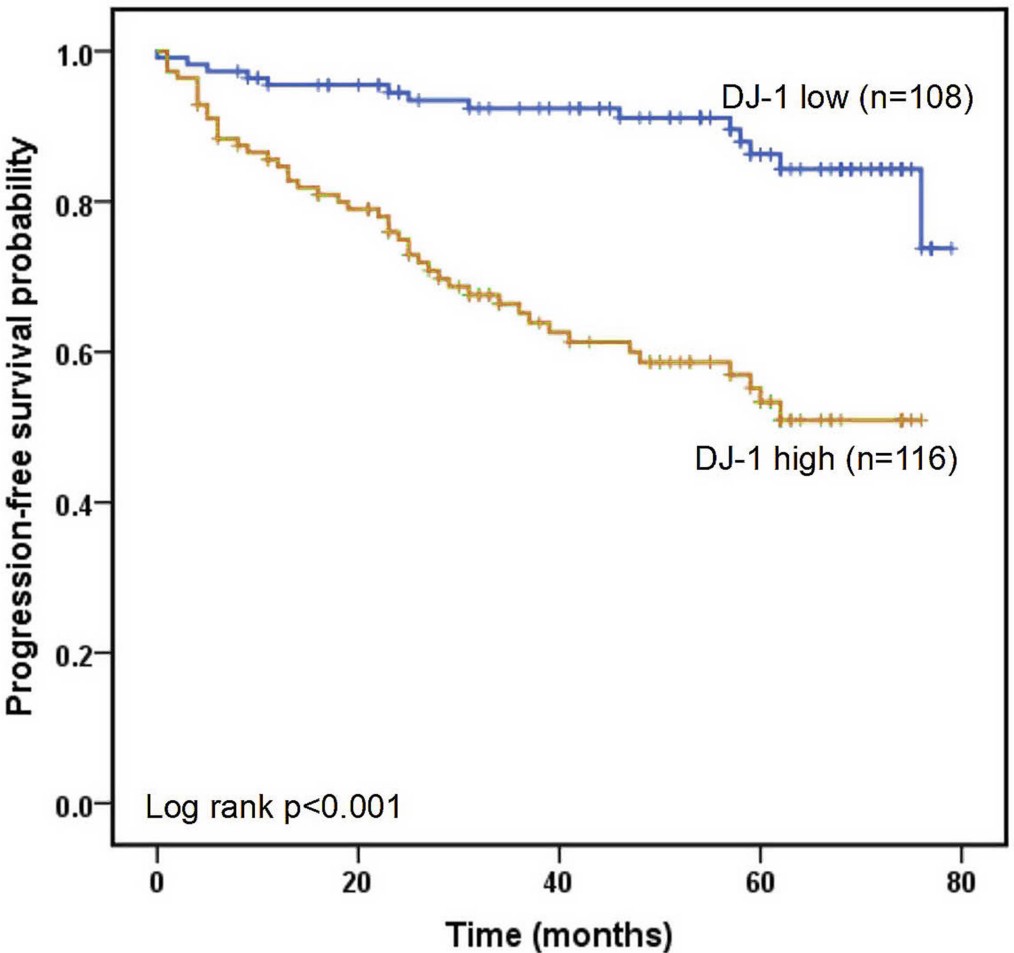

**Figure 5** **Kaplan–Meier analysis of progression-free survival.** Progression-free survival is defined as the time from randomization to radiographic progression or death from any cause.

LUAD. The AUC of DJ-1 was 0.726 with a cutoff value of 4.99 ng/mL for predicting PFS. Univariate and multivariate analyses confirmed that preoperative serum DJ-1 level, tumor stage and smoking history were independent prognostic factors of PFS. These data suggest that serum DJ-1 might be a novel predictor for LUAD.

In addition to the role of DJ-1 in neurodegenerative diseases, different studies point to DJ-1 as an oncogene that was mostly in association with other oncogenes such as c-Myc or H-Ras. In addition, it can act, for example, as a PTEN repressor causing cell proliferation in NSCLC as well as other cancers. DJ-1 is overexpressed in lung cancer (*Han et al., 2017*) and is also secreted by cancer cells and has also been proposed as a cancer biomarker (*Naour et al., 2001*; *Melle et al., 2007*; *Tsuboi et al., 2008*). In this study, we confirmed the overexpression of DJ-1 in LUAD cell lines and serum, which is the most common type of NSCLC. These results corroborated the potential of DJ-1 as a biomarker for LUAD.

**Table 4  Univariate and multivariate analyses of prognostic factors of PFS.**

| Variable | Univariate analysis | | | Multivariate analysis | | |
|---|---|---|---|---|---|---|
| | HR | 95% CI | *P* | HR | 95% CI | *P* |
| Gender (male) | 0.519 | 0.311–0.868 | 0.012 | 1.079 | 0.560–2.081 | 0.820 |
| Age >60 | 1.182 | 0.708–1.971 | 0.522 | | | |
| Tumor size >2 cm | 2.039 | 1.210–3.435 | 0.007 | 1.236 | 0.641–2.380 | 0.527 |
| Tumor number (multiple) | 1.085 | 0.549–2.142 | 0.815 | | | |
| Tumor stage (advanced) | 3.255 | 1.894–5.592 | <0.001 | 3.089 | 1.785–5.346 | <0.001 |
| Lymph node metastasis | 2.393 | 1.291–4.435 | 0.006 | 0.567 | 0.216–1.487 | 0.248 |
| Distant metastasis | 2.531 | 0.916–6.992 | 0.073 | | | |
| Differentiation | | | | | | |
|     moderate vs. well | 2.321 | 1.078–4.998 | 0.031 | 2.133 | 0.953–4.774 | 0.065 |
|     poor vs. well | 3.422 | 1.601–7.312 | <0.001 | 1.949 | 0.751–5.060 | 0.170 |
| Smoking | 2.497 | 1.417–4.402 | 0.002 | 1.820 | 1.021–3.244 | 0.042 |
| Hypertension | 0.800 | 0.414–1.545 | 0.506 | | | |
| Diabetes mellitus | 0.749 | 0.299–1.872 | 0.536 | | | |
| DJ-1 (>4.99 ng/mL) | 5.696 | 2.933–11.059 | <0.001 | 5.298 | 2.697–10.406 | <0.001 |
| CEA (>4.3 ng/mL) | 1.252 | 0.663–2.365 | 0.488 | | | |
| CYFRA21-1 (>3.3 ng/mL) | 1.178 | 0.535–2.594 | 0.685 | | | |
| NSE (>16.3 ng/mL) | 1.645 | 0.986–2.747 | 0.057 | | | |

In this study, our result showed that serum DJ-1 was significantly higher in males than females which previous studies have never reported. It may be attributed to differences in sample size. DJ-1 expression was also correlated with smoking history and lymph node metastasis in LUAD patients. Several studies demonstrated that later stage NSCLC patients had a significantly higher level of serum DJ-1 than those with early-stage cancer (*Fan et al., 2016*; *Kim et al., 2005*). However, *Han et al. (2017)* found that the DJ-1 expression level was higher in stage I than in stage II–IV lung cancer, which may be attributed to different study populations. Additionally, our findings conflict with the results of lower DJ-1 levels in lymph node metastasis from *Han et al. (2017)*. Another study showed that DJ-1 levels were slightly higher in pancreatic cancer patients with lymph node metastasis than in those without metastasis, although the differences did not reach statistical significance (*He et al., 2011*), which agrees with our study.

CEA, CYFRA21-1 and NSE are routine tumor markers of lung cancer, which are not sensitive or specific enough for a reliable evaluation. As a result, numerous recent studies have been performed to look for new diagnostic markers. In our study, we evaluated and compared the diagnostic performance of DJ-1, CEA, CYFRA21-1 and NSE in LUAD. The results revealed an AUC of 0.758 with a sensitivity of 63.8% and a specificity of 78.6% for DJ-1, which showed the best diagnostic value of all markers for discriminating LUAD *versus* the controls. These results suggest that serum DJ-1 may be a diagnostic biomarker for LUAD.

Moreover, a ROC curve analysis for predicting PFS indicated that DJ-1 was superior to other biomarkers. The results of the Kaplan–Meier analysis indicated that LUAD patients

with high DJ-1 levels had shorter PFS than those with lower levels. Therefore, an increase in serum DJ-1 levels is an indication of poor survival. Serum tumor biomarkers can be used as prognostic indicators in LUAD in clinical application (*Ardizzoni et al., 2006*; *Holdenrieder et al., 2017*). *Dal Bello et al. (2019)* revealed that CEA or CYFRA21-1 may serve as a reliable early marker of efficacy that is significantly associated with better DCR and PFS after treatment with nivolumab, and NSE was not significant for monitoring the efficacy of nivolumab. A serum CYFRA21-1 level $\geq$ 2.2 ng/mL was an independent predictor of a favorable PFS (*Shirasu et al., 2018*), while according to other authors (*Kataoka et al., 2018*), a baseline serum CEA level $\geq$ 5 ng/mL was associated with a worse PFS. Elevated serum CYFRA 21-1 was associated with shorter PFS and OS in patients with NSCLC treated with EGFR-TKIs, and serum CYFRA 21-1 may be useful in helping determine the appropriate use of EGFR-TKI therapy in patients with NSCLC. CEA was not a prognostic factor in people with a high burden of lung cancer caused by smoking, nor it was related to PFS or OS (*Takeuchi et al., 2017*). In our present study, there was no significant difference in survival time between patients with different levels of CEA and CYFRA21-1 levels except NSE. These results demonstrate that DJ-1 is more significant than other traditional tumor markers in predicting PFS. Subsequently, univariate and multivariate analyses showed that serum DJ-1 levels were an independent prognostic factor in LUAD patients. Thus, serum DJ-1 could also be utilized as a potential prognostic predictor of LUAD.

There are some limitations in our study. First, this is a single-center study with a small sample size, which may cause deviation. Overall, out findings need to be validated on a larger scale. Second, our study only included three routine tumor markers for comparison, and some other markers were not included such as SCCA and miRNAs.

## CONCLUSIONS

In conclusion, our study is the first to demonstrate the clinical value of DJ-1 in LUAD. DJ-1 is significantly upregulated in LUAD cells. Compared to traditional biomarkers, DJ-1 shows better diagnostic efficiency. Furthermore, serum DJ-1 is significantly and independently associated with PFS. The above results prove that DJ-1 may serve as a novel biomarker for the diagnosis and prognosis of LUAD.

## ACKNOWLEDGEMENTS

The authors would like to acknowledge all study participants and collaborators.

### Funding

This research was supported by the National Natural Science Foundation of China (grant number: 82102488). The funders had no role in study design, data collection and analysis, decision to publish, or preparation of the manuscript.

## Grant Disclosures

The following grant information was disclosed by the authors:
National Natural Science Foundation of China: 82102488.

## Competing Interests

The authors declare there are no competing interests.

## Author Contributions

- Lin Wang performed the experiments, analyzed the data, prepared figures and/or tables, authored or reviewed drafts of the article, and approved the final draft.
- Li Wei performed the experiments, prepared figures and/or tables, and approved the final draft.
- Shuxian Miao analyzed the data, prepared figures and/or tables, and approved the final draft.
- Wei Zhang conceived and designed the experiments, authored or reviewed drafts of the article, and approved the final draft.

## Data Availability

 The raw measurements are available in the Supplementary File.

## Supplemental Information

Supplemental information for this article can be found online at http://dx.doi.org/10.7717/peerj.16845#supplemental-information.

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
