# Peer review of "Clinical value of serum DJ-1 in lung adenocarcinoma"

_PeerJ, doi:10.7717/peerj.16845_

## Round 0.1 · original submission · Major Revisions

The PeerJ manuscript 2023:07:88698:0:1 titled "Clinical value of serum DJ-1 in lung adenocarcinoma" has undergone a thorough review process, and there is a consensus among the reviewers that major revisions are required to strengthen the study's overall scientific rigor and clarity.

One primary concern raised by the reviewers is the need for greater coherence in articulating the study's objectives. There appears to be an inconsistency in the stated aims between the introduction and conclusion sections, particularly in terms of whether the focus is on diagnostics, monitoring progression, or prognostics. Aligning these objectives throughout the manuscript will enhance the overall clarity and coherence of the study.

Another critical area that requires attention is the source of serum DJ-1. Reviewers have highlighted the importance of providing direct evidence regarding whether serum DJ-1 is primarily secreted by lung adenocarcinoma (LUAD) cells or if it originates from other cell types. This information is pivotal in supporting the conclusions drawn in the study and establishing the clinical relevance of DJ-1 as a biomarker for LUAD.

Furthermore, the manuscript should clearly articulate its unique contribution in light of prior research. A previous study (Fun et al., 2016) has reported similar findings, and it is imperative to demonstrate how this study adds distinctive value to the existing body of knowledge. This clarification will help establish the novel significance of the study's findings.

Additionally, addressing the statistical analysis is crucial. The manuscript should include a dedicated section that provides transparency on the methods used for data analysis. Referencing alternative approaches, such as Bayesian shrinkage prior models, as suggested by one of the reviewers, may also be beneficial in validating the study's findings.

Moreover, clear explanations for the choice of sample size and detailed inclusion/exclusion criteria are essential to enhance the study's rigor and reproducibility. Providing this information will offer transparency regarding the selection of participants and strengthen the validity of the study's conclusions.
Lastly, improving the clarity and legibility of graphs and tables is recommended to enhance the presentation of results. Clear visual representation is pivotal in facilitating the reader's understanding and interpretation of the data.

In summary, carefully addressing the comments and concerns of the reviewers through substantial revisions will be paramount for the manuscript to meet the necessary standards for publication. Clarifying the study's objectives, providing direct evidence for the source of serum DJ-1, demonstrating the unique contribution of this study, and improving the presentation of results are critical areas for improvement. Additionally, providing detailed information on statistical analysis, sample size determination, and inclusion/exclusion criteria will strengthen the overall scientific rigor of the study.

Reviewer 1 ·

Basic reporting

This study aims to identify clinical value of dj1 as a predictor of lung adenocarcinoma. However, the paper has an ambiguous and inconsistency aim between introduction and conclusion between diagnostics, monitor progression, prognostics.

Experimental design

No comment

Validity of the findings

1. Since DJ1 is also secreted in other cancer and the exclusion was excluding history of other cancer, how do you make sure that the increase of dj1 is caused by lung adenocarcinoma?
2. Is there potential bias in those with Parkinson disease?
3. Due to inconsistency in the basic reporting, the authors seem to confuse what to report diagnostic accuracy? ppv? npv? the study has shown that low sensitivity and low specificity for the DJ1.
any comments from the authors?

Cite this review as

Reviewer 2 ·

Basic reporting

This study showed that serum levels of DJ-1 protein were higher in lung adenocarcinoma (LUAD) in 224 cases compared to benign pulmonary disease group. And univariate and multivariate analyses revealed that preoperative serum DJ-1 level was an independent prognostic factor of PFS in LUAD cases.

Experimental design

This study lucks direct evidence which serum DJ-1 is secreted by LUAD cells. Authors are strongly recommended to add new data which DJ-1 in the serum is in part derived from LUAD cells. The data should strengthen the conclusion of this study. If the authors consider that DJ-1 in the serum is mainly derived from other cells not from cancer cells, please identify the non-cancer cell.

Validity of the findings

Although authors concluded that the clinical value of serum DJ-1 as a novel predictor for LUAD, similar results were obtained by Fun et al 2016 in 300 lung cancer patients of which about half of cases were LUADs (n=151). Authors appropriately sited this paper at ref. [12] and described that confirmed the high levels of serum DJ-1 in discussion. Thus, novel value of serum DJ-1 in LUAD is limited.

Cite this review as

Reviewer 3 ·

Basic reporting

The manuscript entitled "Clinical value of serum DJ-1 in lung adenocarcinoma" found that serum DJ-1 levels were significantly higher in the lung adenocarcinoma group compared to the benign pulmonary disease group and healthy controls. This study proposes that serum DJ-1 appears to be a promising diagnosticbiomarker and prognostic predictor for lung adenocarcinoma. Interestingly, the past few years have seen the emergence of DJ-1 as a significant biomarker of a number of other systemic malignancies besides lung adenocarcinoma. This research studied the role of DJ-1 in lung adenocarcinoma for the first time.
In the introduction section, the role of DJ-1 as an important biomarker in many systemic malignancies should be cited. It can better demonstrate the rationality and importance of this study.

Experimental design

no comment

Validity of the findings

Most of their data seem to be convincing. In the table 4, the p value of tumor stage in the univariate analyses for PFS is 0.000, check it please.

Additional comments

In discussion sectin, the line 189 to 192. It is contradictory that quoting the increased DJ-1 caused by androgen deprivation therapy in prostate cancer to explain that the higher DJ1 in male patients compared to female patients is due to high androgen levels. In this manuscript, DJ-1 levels were associated with smoking history. The higher DJ-1 levels in male patients compared to female patients may be due to the higher proportion of smoking history in male patients in the statistical data.

Cite this review as

Reviewer 4 ·

Basic reporting

Kindly advise on the inclusion exclusion criteria. How was the sample size chosen?
Write a section on the statistical analysis part.
Add this reference and summarize regarding alternative ways to conduct the analysis. Bhattacharyya et, al. Applications of Bayesian shrinkage prior models in clinical research with categorical responses.

Experimental design

What were the outcomes and features involved in the study ?

Validity of the findings

Address with legible graphs and tables

Cite this review as

---

## Round 0.2 · accepted · Accept

The reviewers do not have any new concerns about the revised version of the manuscript and I am happy to accept the manuscript.

Reviewer 4 ·

Basic reporting

Comments addressed

Experimental design

Comments addressed

Validity of the findings

Comments addressed

Cite this review as